# Learning complexity of many-body quantum sign structures through the lens of Boolean Fourier analysis

Ilya Schurov[1,2], Anna Kravchenko[1], Mikhail I. Katsnelson[1], Andrey A. Bagrov[*1], Tom Westerhout[1]

[1] Institute for Molecules and Materials, Radboud University, Heyendaalseweg 135, 6525AJ Nijmegen, The Netherlands

[2] Omnifold Inc.

\* andrey.bagrov@ru.nl

September 11, 2025

## Abstract

We study sign structures of the ground states of spin-$1/2$ magnetic systems using the methods of Boolean Fourier analysis. Previously it was shown that the sign structures of frustrated systems are of complex nature: specifically, neural networks of popular architectures lack the generalization ability necessary to effectively reconstruct sign structures in supervised learning settings. This is believed to be an obstacle for applications of neural quantum states to frustrated systems. In the present work, we develop an alternative language for the analysis of sign structures based on representing them as polynomial functions defined on the Boolean hypercube – an approach called Boolean Fourier analysis. We discuss the relations between the properties of the Boolean Fourier series and the learning complexity of sign structures, and demonstrate that such polynomials can potentially serve as variational ansätze for the complex sign structures that dramatically outperform neural networks in terms of generalization ability. While ansätze of this type cannot yet be directly used in the context of variational optimization, they indicate that the complexity of sign structures is not an insurmountable curse, and can potentially be learned with better designed NQS architectures. Finally, we show how augmenting data with Boolean functions can aid sign prediction by neural networks.

# 1  Introduction

Understanding the ground state properties of strongly correlated many-body quantum systems – such as the Heisenberg or Hubbard models – remains one of the central challenges in modern condensed matter theory [1, 2]. The wave functions of such systems typically exhibit extensive quantum entanglement, leading to complex patterns in the distribution of quantum mechanical phases (or, in case of real-valued Hamiltonians, signs) [3, 4] when expanded in computational bases. These patterns encode fundamental information about the quantum state and play a crucial role in determining both the physical properties of the system [5] and the computational complexity of studying it[1] [9]. For a real-valued Hamiltonian, it is always possible to choose a basis in which its ground state (and eigenstates in general) is real, and the phase pattern reduces to a sign structure $S$. For spin-1/2 lattice systems with two-dimensional local Hilbert spaces, we can write:

$$|\psi_{\mathrm{GS}}\rangle = \sum_{j=1}^{2^N} |\psi_j| S_j |j\rangle, \tag{1}$$

$$S : |j\rangle \to S_j \in \{-1, +1\},$$

where $N$ is the number of lattice sites, $|j\rangle$ vectors form the computational basis of the problem (usually these are eigenstates of the $\sigma_z^{\otimes N}$ operator), and $|\psi_j|$ are the corresponding absolute values of the wave function coefficients.

While non-trivial signs (or phases) of quantum states are the primary feature that distinguishes quantum mechanics and wave functions from classical statistical mechanics and probability distributions [10, 11] and makes the former much more complex to study, the non-trivial question becomes how to quantify this complexity. The main endeavor of this paper is to develop a language for the formal characterization of sign structures that can be used to explain in a formal way why some systems are more amenable to certain computational methods while others prove more challenging. There is no single definition of complexity, and the choice of the relevant one is a complex problem by itself [12, 13]. In this paper, we apply two general notions of complexity to the sign structures of quantum wavefunctions and study the relations between them.

The first notion is *description complexity*: it addresses the question of what is the length of the shortest description that allows reconstruction of an object, at least approximately. This assumes the description is made in some formal and unambiguous language, and the choice of language is crucial—different languages lead to different complexity notions. If the language is Turing-complete and no approximations are allowed, description complexity becomes Kolmogorov complexity, for which a rich theory exists [14]. Unfortunately, Kolmogorov complexity is uncomputable for any given object [15, 16]. We therefore need to choose a more restrictive language for this notion to be practically applicable.

---

[1]Here we focus on sign structures of many-body wavefunctions that are not directly related to the infamous quantum Monte Carlo sign problem [6–8], but we do believe that there is a deep connection between these concepts.

The second notion is *learning complexity*, which addresses how difficult it is to learn a rule describing an object, at least approximately, by observing only a small part of it. We treat the object as a dataset for a supervised learning task and interpret the performance of a machine learning algorithm (e.g., decision trees or neural networks) as a complexity measure – better performance indicates lower complexity. To make this definition complete, we must specify the learning algorithm, and, as we show below, this choice can be crucial.

As both definitions attempt to capture the same underlying notion, one can expect them to be aligned with each other. Indeed, if an object has a simple description (e.g., it is a linear function plus Gaussian noise), it can presumably be efficiently learned (e.g., with linear regression), and vice versa. However, counterexamples exist: a long sequence of numbers produced by a good deterministic pseudorandom number generator can be described via a short instruction (the generator code and its initial state) [17] and at the same time be extremely difficult to learn [18]. Thus, both definitions grasp different aspects of complexity, making it interesting to study their relationship in specific contexts.

Both notions of complexity are relevant to the physical applications. The description complexity basically asks for the simplest model (selected from some class of models) that can describe an object, which is a core question of any physical research. The relevance of the learning complexity becomes evident with the development of machine learning methods for physical applications, such as neural quantum states (NQS) [19], which have become the most widely adopted approach for learning quantum many-body states.

The learning complexity of ground states in the context of NQS has been already studied from the practical point of view. Neural networks of popular architectures have been shown to struggle with learning sign structures of ground states in highly frustrated magnetic systems [20, 21], and similar challenges can be expected to arise in fermionic systems where complex sign structures are ubiquitous due to antisymmetry requirements. While extensive efforts have focused on improving algorithms at the operational level – through better variational ansätze and optimization schemes [22, 23] – much less is understood about the fundamental structural properties that make sign structures inherently complex [9].

It suggests that the field would benefit from a unifying framework that can provide common ground for both description and learning complexities of ground states and allow to study them on a conceptual level. In the present paper we introduce such a framework by applying theory of Boolean Fourier analysis to the sign-structures of ground states of quantum many-body systems.

Our approach extends the discussion of complexity of ground states in two directions. First, Boolean Fourier analysis allows us to consider not only the learning complexity but also the description complexity of sign structures – a perspective completely missed previously. Second, it enables construction of new learning algorithms that yield different flavors of learning complexity.

As a testbed, we consider the sign strucutres of spin-1/2 Heisenberg antiferromagnet ground state on frustrated lattices. To implement our approach, we view the sign structures as Boolean functions on an $N$-dimensional hypercube $B^N$:

$$S : B^N \equiv \{-1, +1\}^N \to \{-1, +1\}, \tag{2}$$

which is possible since vectors of the computational basis can be viewed as binary sequences of length $N$: $|\uparrow\downarrow\downarrow\uparrow\ldots\rangle = |+1, -1, -1, +1\ldots\rangle$. We then decompose $\mathcal{S}$ in a basis of *parity functions* acting on the binary strings representing vectors $|j\rangle$. To this end, we consider

Hilbert space $\mathcal{F}_N$ of functions $f : \sigma \in B^N \to \mathbb{R}$ with the following scalar product:

$$\langle f \mid g \rangle = \frac{1}{2^N} \sum_{\sigma \in B^N} f(\sigma) g(\sigma). \tag{3}$$

We call elements of $\mathcal{F}_N$ *signals*. For any subset $I = \{i_1, i_2, \ldots, i_k\} \subset \{1, 2, \ldots, N\}$, denote

$$\chi_I(\sigma) = \sigma_{i_1} \ldots \sigma_{i_k}. \tag{4}$$

Functions $\chi_I$ are called *parity functions*[2]. In quantum mechanical terms, each parity function is associated with a subset of lattice sites and, for each basis vector, computes the product of spin projections on this subset and returns a sign-valued output (throughout this paper, we normalize spin projections to $\pm 1$, not $\pm 1/2$). Hence, a single parity function can be viewed as the most primitive non-trivial example of sign structure. It is easy to show that the collection of parity functions $(\chi_I : I \subset \{1, 2, \ldots, N\})$ is an orthonormal basis in $\mathcal{F}_N$. Thus any signal $f \in \mathcal{F}_N$ can be uniquely represented as a linear combination of parity functions

$$f = \sum_{I \subset \{1, \ldots, N\}} a_I \chi_I, \tag{5}$$

where numbers $a_I \in \mathbb{R}$, called Boolean Fourier coefficients of $f$, can be determined as

$$a_I = \langle \chi_I \mid f \rangle. \tag{6}$$

The set of all Boolean Fourier coefficients is called power spectrum.

We will study the complexity of sign structures in terms of their Boolean Fourier series. Our motivation for taking this approach is two-fold. First, parity functions (related to *exclusive OR* or *XOR* operations in computer science) are simple functions that produce linearly non-separable data sets, and the task of classifying *XOR* outputs is often viewed as the most primitive problem requiring neural networks with non-linear activations. Hence, when analyzing sign structures in the context of NQS training, it is natural to decompose them as series of parities and relate their learning complexity to the properties of these series. Second, as we will show in Section 2.1, the simplest non-trivial sign structure of the Heisenberg spin-1/2 antiferromagnet on a square lattice – the Marshall-Peierls sign rule – is itself a single parity function, i.e., the simplest Boolean Fourier series. This makes it natural to represent more complex sign structures in the same framework and use Boolean Fourier analysis as a common language for analyzing their complexities and relating them to the physical properties of the corresponding systems. This framework enables us to construct complexity measures for sign structures that are independent of specific neural network architectures or optimization procedures. Moreover, this language can be exploited to develop surprisingly effective learning procedures for sign structures and to improve existing NQS-based ansätze.

To cover a diverse variety of geometries, we consider ground states of the spin-1/2 Heisenberg antiferromagnet on square, triangular, and kagome lattices. Since we aim to understand the intrinsic features of their sign structures explicitly, we consider relatively small clusters (up to 24 spins), which allows us to obtain numerically exact solutions and to perform exact Boolean Fourier transformations on the complete Hilbert space basis. The cluster geometries are shown in Fig. 1. We wish to continuously tune frustration between simple (ordered) and complex (e.g., spin liquid) regimes. For the square lattice, this can be achieved by adding next-nearest-neighbor exchange $J_2$, with $J_2 = 0$ corresponding to trivial antiferromagnetic order with the Marshall-Peierls sign rule [24], $J_2/J_1 \simeq 0.55$ that

---

[2]If we recode $-1$ as 1 and 1 as 0, parity function becomes XOR or summation modulo 2.

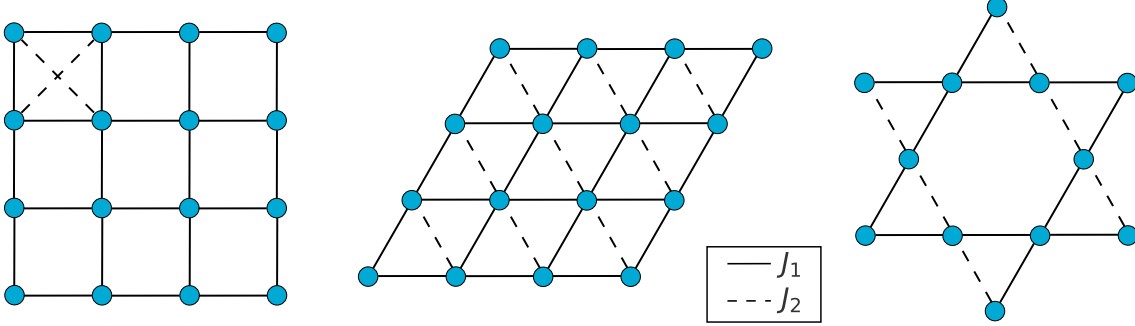

Figure 1: Three antiferromagnetic Heisenberg models considered in the paper: square lattice with next-nearest exchange, triangular, and kagome with spatial anisotropy. The value of $J_2/J_1$ controls the level of frustrations, with $J_2/J_1 = 0$ corresponding to bipartite lattices.

might correspond to a spin liquid [25, 26], and $J_2/J_1 \gg 1$ yielding simple stripe order with a modified Marshall-Peierls rule. For the triangular and kagome lattices, we introduce spatial anisotropy with tunable exchange couplings along one direction. The Hamiltonian takes the form

$$\hat{H} = J_1 \sum_{\langle a,b \rangle} \vec{\sigma}_a \otimes \vec{\sigma}_b + J_2 \sum_{\langle\langle a,b \rangle\rangle} \vec{\sigma}_a \otimes \vec{\sigma}_b, \tag{7}$$

where, for each lattice geometry, $\langle a, b \rangle$ denotes links of the unfrustrated sublattice, and $\langle\langle a, b \rangle\rangle$ denotes the links that add frustration. For every choice of geometry and couplings, we construct the numerically exact ground state using exact diagonalization implemented in the *lattice-symmetries* package [27] and decompose the corresponding sign structure in the Fourier series, which is then analyzed. To better understand critical aspects of sign structure anatomy, we also consider artificially constructed sign structures obtained as deformations of the ground state sign structures.

Here are our main results:

1. We calculate Boolean Fourier series for sign structures of the considered systems for various levels of frustrations and study their properties. Specifically, we introduce several complexity measures for these series, based on the ideas of reconstruction complexity, and demonstrate their physical relevance.

2. We study the relation between the learning complexity of sign structures in the context of neural networks and the reconstruction complexity measures we introduced. We show that while there is indeed a clear correlation between the reconstruction and learning complexities, the latter cannot be fully reduced to the former.

3. We develop new learning procedures that allow reconstruction of sign structures from small samples of basis elements. These procedures are represented by explicit equations that relate the predictions with the training set. They do not require any opaque optimization-based learning phases and perform surprisingly well. For frustrated systems, they outperform previous neural network results by an order of magnitude in terms of generalization ability. This sheds a new light on the notion of learning complexity in the context of sign structures and demonstrate the importance of the choice of the learning algorithm.

4. Finally, as a practical outcome, we demonstrate that augmenting neural networks with additional inputs calculated as parity functions significantly improves the quality of NQS approximations.

The paper is organized as follows. In Section 2, we discuss properties of the Fourier series of sign structures and show how they correlate with learning complexity. In Section 3, we introduce the concepts of Hadamard (Section 3.1) and Fourier learning (Section 3.2) – formal approaches to reconstruct complex sign structures from small datasets in a supervised manner without involving neural networks. Finally, in Section 3.3, we show how augmenting data with outputs of parity functions can aid neural networks in learning signs.

## 2 Fourier series of sign structures

Let $f$ be a Fourier series decomposition of a certain ground state sign structure:

$$f = \sum_i a_i \chi_{I_i}, I_i \subset \{1, \ldots, N\} \tag{8}$$

Assume that terms in the sum are sorted in descending order: $|a_i| \geq |a_{i+1}|$. In each specific case, this object can be characterized by several features. We will mainly focus on the following aspects:

- What is the fall-off of the series of sorted coefficients $\{a_i\}$?

- What geometric structures do the dominant parity subsets have (i.e., what is the shape of spin subsets $I_i$ corresponding to the largest $a_i$ coefficients)?

- What are the signs of $a_i$ in the Fourier expansion? In other words, how does the sign structure change when we transform it from the original computational basis to the basis of parity functions?

Intuitively, the answer to the first question should depend on the degree of frustration. The difficult-to-learn complicated sign structures near the spin liquid phase probably cannot be reduced to just a few terms in the Fourier expansion, implying slow fall-off of the coefficients, while sign structures of simple magnetic orders, like the Marshall-Peierls rule, are well-described by a handful of the most relevant parity functions. In what follows, we will demonstrate that the number of relevant terms in the expansion indeed correlates very well with the learning complexity of the sign structure.

The second question is most natural to consider in the context of machine learning and NQS. When variational NQS are used to learn sign structures, they solve a binary classification problem, assigning each Hilbert space basis vector to either the "+1" or "-1" class based on features they identify in the pattern of up and down spins on the lattice. Each parity function can then be regarded as an individual feature. Again, in the simple case of the Marshall-Peierls rule, the only feature that needs to be identified for successful sign prediction is the product of spins on one of the two sublattices of a bipartite lattice. If many parity functions contribute to the sign structure, they can lead to conflicting sign predictions. In simple terms: if, based solely on feature 1 (given by some parity function $\chi_{I_1}$), the NQS tends to assign a positive sign to some basis vector, while based on feature 2 (given by parity function $\chi_{I_2}$), it tends to assign a negative sign to the same vector, it could be challenging to learn a combined sign rule that incorporates both features. The degree of this conflict depends on the geometric arrangement of the parity functions. In

what follows, we will show that sometimes it is easier to learn a sign structure defined by $\sim 10^4$ parity functions than a structure defined by $\sim 10^2$ parity functions.

Finally, it is also interesting to examine the signs of the expansion coefficients in Eq. (8), which can be regarded as a "secondary" sign structure. On one hand, this secondary structure does not have obvious meaning of its own, since signs of $a_i$ coefficients do not directly correspond to signs of the wave function in the computational basis. At the same time, we will see that these secondary sign structures exhibit non-trivial patterns, and their complexity correlates with the learning complexity of the original sign structures.

## 2.1 Fall-off of the Fourier series

First, we pursue the simple idea that easily learnable sign structures should have short Boolean Fourier representations, i.e., the $\{|a_i|\}$ series decays fast enough. This idea is grounded in classical results from computational learning theory. In [28], a fundamental connection between the power spectrum structure of Boolean functions and their learnability was established: functions that can be represented with short Fourier series also have short Disjunctive Normal Form representations, which, in simple terms, means that they are defined by a small number of elementary rules that are not difficult to learn.

To begin with, let us consider the simplest non-trivial sign structure: the Marshall-Peierls rule describing signs of the ground state of the quantum antiferromagnetic Heisenberg model on a bipartite lattice. It states that the sign of a wave function coefficient in front of a basis element is determined by the parity of magnetization of one of the two sublattices (denoted here as $A$):

$$S_i = \prod_{k \in A} \sigma_i^k, \ \ \sigma_i^k = \pm 1, \tag{9}$$

where $\sigma_i^k$ is the binary encoding for spin "up" or spin "down" at position $k$ of basis vector $i$. This rule can be represented as exactly one parity function $S = \chi_{I_{MP}}$, where $I_{MP}$ is the full set of indices of sublattice $A$. The Marshall-Peierls rule is known to be easily deduced by a simple neural network without any prior knowledge [20].

Before we proceed to more complex cases, one comment is in order. The ground state of the bipartite Heisenberg antiferromagnet belongs to the sector of minimal magnetization, and all amplitudes of basis vectors with non-minimal magnetization $|m_i| = \left|\sum_k \sigma_i^k\right|$ are zero. Still, the Marshall-Peierls rule expressed as a single parity function formally takes $+1$ or $-1$ values on them, which does not make much sense. For us, it will be convenient if sign rules have meaning on the complete Boolean hypercube and return precisely zero values for basis vectors outside the minimal magnetization sector. In particular, for the Marshall-Peierls rule modified in this way, its expansion in the parity function basis would have more than one non-zero term (though, all terms except of two will have very small coefficients).

Two complexity scores can readily be defined based on the asymptotic behavior of Eq. (8). One is the *participation ratio (PR)*:

$$PR \equiv \frac{(\sum_i a_{I_i}^2)^2}{\sum_i a_{I_i}^4}, \tag{10}$$

which is the inverse of the *inverse participation ratio (IPR)*, a quantity commonly used [29] to measure localization of quantum states. Here we apply it not to quantum states in the computational basis, but to the Fourier spectrum of sign structures. If there is only one non-zero term in the Fourier expansion, like for the single-parity Marshall-Peierls rule, its $PR$ equals 1. The more delocalized the expansion is, the larger its $PR$ becomes, with

maximum value $C_n^{[n/2]}$ for minimal magnetization ground states and $2^N$ for generic sign structures. As we will see shortly, $PR$ can be naturally used as a complexity measure for sign structures (as well as general signals on the hypercube).

Another possible measure of complexity is *weight complexity*. Given a signal Fourier expansion $\sum_i a_{I_i} \chi_{I_i}$, the Fourier weight associated with subset $I_i$ can be introduced as

$$w_{I_i} \equiv \frac{a_{I_i}^2}{\sum_j a_{I_j}^2}. \tag{11}$$

In most practical cases, it is sufficient to have a good approximation to the exact sign structure, which can be achieved by retaining only $K$ main terms of the expansion bearing a fraction $p$ of the spectral weight:

$$\mathcal{C}_W(p) \equiv \min K : \ \sum_{i=1}^{K} w_{I_i} \geq p. \tag{12}$$

As larger values of $K$ mean a larger number of parity/XOR primitives to be represented explicitly or learned by a neural network to achieve the desired accuracy, intuitively, the complexity of a sign structure should grow with $K$. Hence, we define $\mathcal{C}_W(p)$ as the threshold-dependent weight complexity of a signal.

To introduce one more measure of complexity based on the properties of the Fourier spectrum, we first need to define a score of similarity between sign structures. One natural metric is the *accuracy score*, which for two sign structures $f$ and $g$ is defined as follows:

$$d_a(f, g) \equiv \frac{\#\{\sigma : f(\sigma) = g(\sigma) \neq 0\}}{\#\{\sigma : f(\sigma) \neq 0 \text{ and } g(\sigma) \neq 0\}}, \tag{13}$$

where $\sigma$ are basis vectors (binary sequences). This accuracy score is designed so that it does not care about sign discrepancy on basis vectors with zero amplitudes. For example, the single-parity Marshall-Peierls rule given by $\chi_{I_{PM}}$ and the modified one are identical according to this metric ($d_a = 1$). However, this metric does not take into account the relative importance of non-zero amplitudes of the wave function and imposes equal penalties for discrepancy in the sign of a basis vector with large amplitude and in the sign of a basis vector with negligibly small but non-zero amplitude. This can be corrected by considering *sign overlap*:

$$d_\psi(f, g) \equiv \frac{\sum f(\sigma) g(\sigma) |\psi(\sigma)|^2}{\sum |\psi(\sigma)|^2} \tag{14}$$

where sign discrepancies are weighted with ground state $\psi$ amplitudes.

Another natural approach to define the complexity of sign structure $f$ is through the complexity of its reconstruction from the corresponding Fourier expansion; hence we call it *reconstruction complexity*. For that, let us employ one of the two similarity scores, fix some tolerance level $\epsilon$, find the Fourier expansion of $f$, and truncate it at the $K$-th term (we dub the truncation $f_K = \sum_{i=1}^{K} a_{I_i} \chi_{I_i}$). Similarly to weight complexity, we define reconstruction complexity as the minimal number of terms to be retained in order to keep the truncated sign structure close to the exact one in accordance with the chosen similarity score:

$$\mathcal{C}_R(\epsilon) \equiv \min K : \ d(f, f_K) \geq 1 - \epsilon, \tag{15}$$

where $d = d_a$ or $d_\psi$. For example, for either of the similarity scores and for any arbitrarily small tolerance window $\epsilon > 0$, the reconstruction complexity of the Marshall-Peierls sign rule (both single-parity and modified versions) is exactly 1. This measure of complexity

is more fine-grained than the previous two, but also more computationally expensive: to find the minimally required number of terms $K$, one has to evaluate the similarity score for a number of truncations (e.g., $K$ can be searched for in a bisectional manner), and each evaluation requires performing an inverse Fourier transform of the truncated series, which is numerically expensive.

In this paper, we will focus on small-scale quantum systems and study all three complexity measures. However, for larger systems, participation ratio and weight complexity are more feasible choices, especially because they can be computed by means of Monte Carlo sampling and do not require finding and storing all $\sim 2^N$ Fourier expansion coefficients, which would be the case when computing reconstruction complexity.

With these complexity measures at hand, we can study sign structures of ground states of the frustrated Heisenberg AFM on different types of lattices. To have an additional class of reference structures, besides considering signs we also analyze binarized amplitudes, which we define in the following way. If $\{|\psi_j|\}$ is the set of wave function amplitudes, and median $|\psi|$ is their median (i.e., such a value that at least half of the amplitudes are not smaller than it and at least half are not larger than it), the binarized amplitude function is

$$\mathcal{A}(|j\rangle) = \text{sign}\left(|\psi_j| - \text{median}\,|\psi|\right). \tag{16}$$

The reason we are interested in this object is that a considerable difference in learning complexity of amplitudes and sign structures has been observed and discussed before [20], and it is tempting to compare them within the chosen framework of Boolean Fourier analysis. However, amplitudes *per se* are not binary functions, and to make the comparison fairer, we perform this binarization procedure.

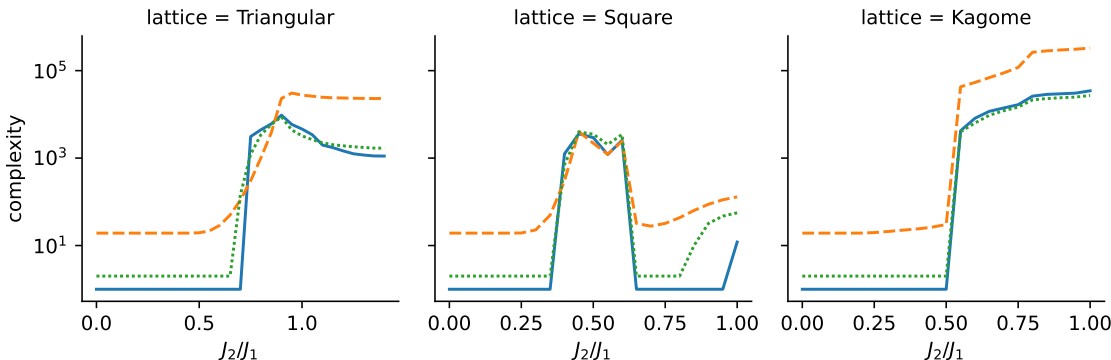

Figure 2: The dependence of different sign structure complexity measures on $J_2/J_1$ ratio in the AFM Heisenberg model. Solid blue line depicts the reconstruction complexity, dotted green line – the weight complexity, and dashed orange line – the participation ratio. The vertical axis is in logarithmic scale.

In Fig. 2, we show the dependence of the complexity measures defined above on $J_2/J_1$ or the ground state sign structures of the Heisenberg AFM on different lattices. The reconstruction complexity shown here is based on the unweighted accuracy score. For the weight complexity, we fix the threshold $p = 0.2$, and for the reconstruction complexity, we set the tolerance level $\epsilon = 0.2$ for the following reasons. In our simulations, we found that the predictions of truncated Fourier series are strongly biased toward better overlap in the sense that achieving an accuracy score of 0.8 usually corresponds to a sign overlap of 0.95 or even larger[3]. On the other hand, we found that the combined weight of Fourier

---

[3]However, since using the accuracy score leads to more stable behavior of $C_R(\epsilon)$ as a function of

expansion terms required to achieve an accuracy score of 0.8 typically lies in the range
$[0.075, 0.25]$, so we chose $p = 0.2$ as the threshold for the weight complexity measure for
better agreement between the two measures.

We see that all three measures behave as expected, developing a considerable increase
in the same regions where neural networks struggle to learn and generalize signs. Namely,
this occurs at $0.4 \leq J_2/J_1 \leq 0.6$ for the square lattice (which is widely believed to be
the vicinity of the spin liquid phase), $J_2/J_1 > 0.8$ for the triangular lattice (where the
spin liquid emerges at $J_2/J_1 > 1.2$), and $J_2/J_1 > 0.5$ for the kagome lattice (also the spin
liquid regime), see Fig. 2 in [20]. The kagome Heisenberg model is known to be the most
difficult for finding a variational approximation to its ground state and sign structure.
And, indeed, in Fig. 2, one can see that, in the frustrated regime, complexity measures
of its sign structure are generally larger than those of the square and triangular lattice
Heisenberg models. The low complexity of the sign structures outside the regime of high
frustrations is easily explainable. For $J_2 = 0$, all three systems obey the Marshall-Peierls
rule and can be effectively represented by just one parity function. Also, it is known that
sign structures of mildly frustrated systems remain close to the Marshall-Peierls rule with
rather high accuracy [21]. However, a closer look at this picture already prompts the
question of whether there is something more behind the learning complexity than simply
the number of relevant terms in the Fourier expansion. For that, one needs to zoom in
on the $J_2/J_1 = 0.52$ point of the kagome system. There, learning complexity is already
extremely high, and variational neural network approximation of the ground state sign
structure is much more challenging than for the square/triangular lattices for any value
of $J_2/J_1$. However, all three complexity measures introduced so far are the same as for
the triangular lattice Heisenberg model at the maximal complexity point. In the next
section, we will analyze what other factors make the kagome antiferromagnet so complex,
and why expansion-length-based measures are not sufficient to fully reflect it. For now,
let us nevertheless focus on the Fourier expansion asymptotics and discuss them in more
detail.

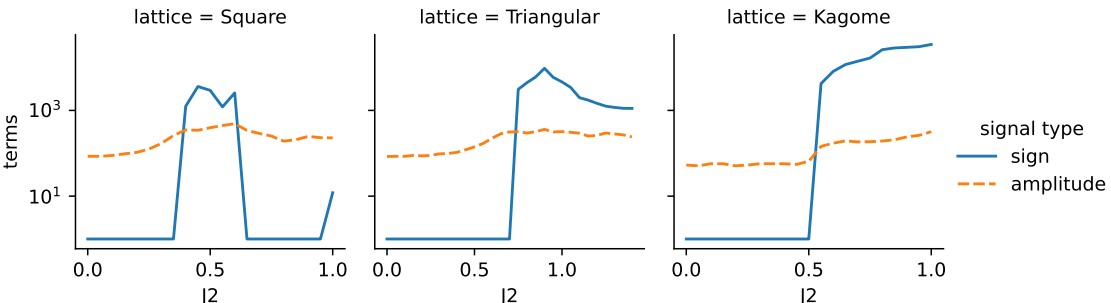

Figure 3: Reconstruction complexity of sign and binarized amplitude. The vertical axis is
in logarithmic scale.

It was shown in [20] that, in the regime of strong frustrations, it is easier for neural
networks to learn amplitudes than signs. Thus it is reasonable to expect that the com-
plexity of amplitudes should be smaller than that of signs. As explained above, to make
the comparison fair, we binarize the amplitude signal. In Fig. 3, we show the reconstruc-
tion complexity of sign structures and binarized amplitudes for the three types of lattices
as a function of $J_2/J_1$ (unweighted accuracy score and tolerance $\epsilon = 0.2$ are adopted).
The reconstruction complexity of binarized amplitudes appears insensitive to the degree

tolerance, we stick to it.

of frustration, being considerably smaller than that of sign structures in the highly frustrated regime, but larger in the non-frustrated regime where signs are trivial. The same result holds for weight complexity. For participation ratio, the complexity of amplitudes is larger than that of signs for the square lattice, but smaller for frustrated triangular and kagome lattices.

Finally, we analyze the scaling of complexities with respect to the system size. In Fig. 4, we show the dependence of reconstruction complexity on the number of spins for signs and binarized amplitudes. We consider frustrated and non-frustrated regimes separately. For each lattice, we fix a pair of values of $J_2$: one that corresponds to strong frustrations, and the other to mild frustrations. One can see that the complexity of binarized amplitudes does not seem to scale with the system size at all in either regime. Moreover, as follows from the bottom panel of the figure, it does not react to frustrations induced by periodic boundary conditions in odd-sized lattices.

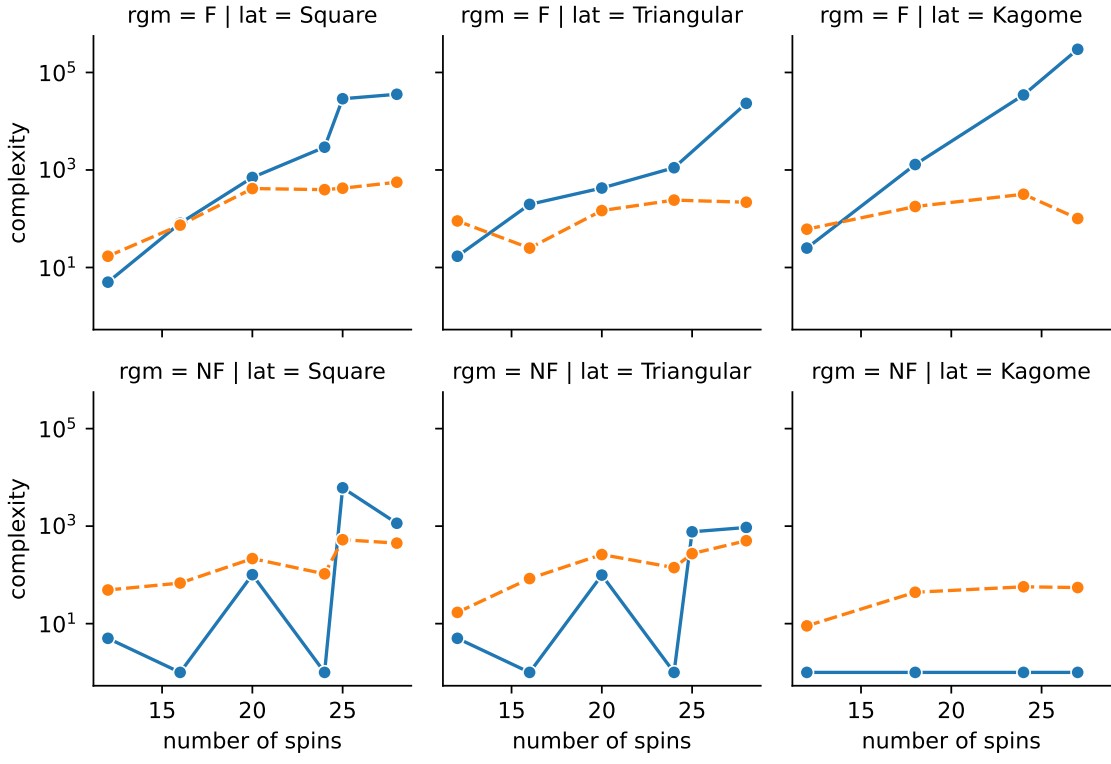

Figure 4: Scaling of the reconstruction complexity measure of signs and amplitudes. Top row: frustrated regime (the values of $J_2/J_1$ are equal to 0.5, 1.4 and 1 for square, triangular, and kagome lattices respectively), bottom row: non-frustrated regime (the corresponding values of $J_2/J_1$ are 0.2, 0.5, and 0.4 respectively). The vertical axis is in logarithmic scale. Solid blue line: sign signal. Dashed orange line: binarized amplitude signal.

## 2.2 Beyond simple Fourier complexity: the role of geometric structure and secondary signs

While the complexity measures introduced in the previous subsection correlate well with neural network learning difficulties, they may not tell the full story. The seminal work of Linial, Mansour, and Nisan [30] showed that the geometric structure of Fourier expansions (specifically, whether the significant terms involve few or many variables – lattice

nodes in our language) affects learnability as much as the total number of terms. This suggests we should examine not just how many parity functions are needed to represent a signal, but also their spatial patterns and geometric arrangements. As we will show, the relationship between Fourier representation of signals and their learnability goes beyond simple coefficient counting.

To investigate this, we perform the following experiment. For a frustrated system, we find the Fourier expansion of its ground state sign structure, then truncate it keeping only a fraction of the largest (by absolute value) coefficients. We use these truncated expansions as learning targets for neural networks. If simple Fourier spectrum-based complexity were the whole story, we would expect that the more terms we keep, the harder the structure becomes to learn.

We train dense feedforward networks (one hidden layer, 512 neurons) on all three families using training sets of $\epsilon_{\text{train}} \cdot C_{N/2}^N$ and test sets of $\epsilon_{\text{test}} \cdot C_{N/2}^N$ vectors from the sector of zero magnetization. Each basis vector $\sigma$ is selected with probability proportional to $|\psi_{GS}(\sigma)|^\alpha$, where $\psi_{GS}$ is the exact ground state obtained by means of exact diagonalization, and $\alpha \geq 0$ is the sampling power. We fix $\alpha = 2$ as the natural quantum mechanical probability weighting. The neural network is trained using Adam optimizer with learning rate $10^{-3}$ over 300 epochs. The loss is binary cross-entropy. The choice of the architecture is motivated by the fact that our new learning methods are physically unbiased, specifically, they do not have any built-in priors like symmetry invariance, so it is natural to compare them with with similarly unbiased neural network. Also, 1-layer dense NN demonstrated better performance than 2-layer dense for frustrated Kagome [20], thus there were no reasons to increase the depth of the network in our experiments.

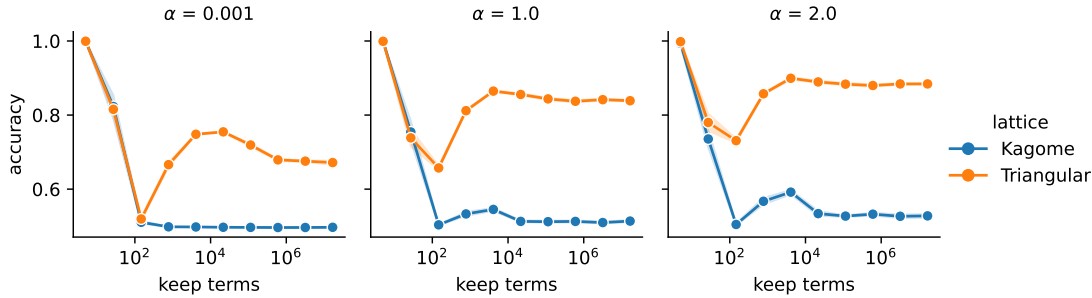

Figure 5: The dependence of learnability on the length of a Fourier expansion. Here $\varepsilon_{\text{train}} = 0.05$, $\varepsilon_{\text{test}} = 0.1$, $J_2/J_1 = 1$ for Kagome lattice and $J_2/J_1 = 1.3$ for triangular lattice. Vertical axis represents accuracy on a training dataset, $\alpha$ is a sampling power for train dataset. The horizontal axis is logarithmic

Fig. 5 shows that this hypothesis does not always hold true: in the case of the Heisenberg AFM on the triangular lattice, the dependence is highly non-monotonic. While learnability does decrease as we add more terms (up to about $10^2$ terms), beyond that point the relationship reverses: adding more terms actually makes the signal easier for neural networks to learn. This suggests that complex sign structures possess some internal consistency that is destroyed by aggressive truncation. When we keep only a few terms, the resulting function lacks the "space" to encode this consistency; when we keep too many terms in the middle range, we capture enough complexity to be challenging but not enough structure to be learnable. This counterintuitive result demonstrates that simple Fourier complexity measures cannot fully explain neural network learnability. The geometric structure and spatial arrangements of the parity functions must play a crucial

role.

Having established that simple complexity measures are not sufficient, we now examine the geometric structure of the parity functions themselves. Parity functions can be viewed as elementary units that capture non-linear interactions between binary variables (spins in our case). This makes it valuable to inspect the spatial patterns of the most influential terms: those with the largest coefficients $|a_I|$ in our Fourier expansions.

In Fig. 14, we visualize the top 7 terms in Fourier expansions of sign and binarized amplitude signals for different values of $J_2/J_1$. Several patterns emerge:

1. For all systems in the unfrustrated regime, the leading term of the sign expansion is Marshall-Peierls.

2. Sign structure expansions consist of high-degree parity functions (involving many spins), while amplitude expansions use mostly low-degree terms (2-4 spins). The exception is the square lattice at large $J_2/J_1$, where Marshall-Peierls-like terms appear in amplitude expansions, but other terms remain low-degree.

3. For the square lattice, sign expansions are dominated by three types of regular sublattices: Marshall-Peierls (checkerboard) for small $J_2$, stripes for large $J_2$ (corresponding to Marshall-Peierls structure on second-nearest-neighbor bonds), and "stretched checkerboard" at intermediate values.

4. For the triangular lattice, both regular and irregular sublattices appear in the frustrated regime. For the kagome lattice, we observe no visually regular sublattices in the frustrated regime.

These observations suggest that it might be easier to learn a sign structure defined by many geometrically regular parity functions (e.g., checkerboard or stripe patterns) than one defined by few parity functions with complex, competing geometries.

To test this hypothesis, we consider ground states of the Heisenberg AFM on square ($J_2/J_1 = 1.0$), triangular ($J_2/J_1 = 0.8$), and kagome ($J_2/J_1 = 1.0$) lattices. For each ground state, we construct three families of truncated sign structures:

1. **Simply truncated**:

$$t_K = \sum_{i=1}^{K} a_{I_i} \chi_{I_i}, \tag{17}$$

preserving the original coefficient magnitudes, with $K = 2^N$ case corresponding to the exact sign structure.

2. **Homogenized**:

$$h_K = \sum_{i=1}^{K} \chi_{I_i}, \tag{18}$$

where all retained parity functions have identical unit coefficients. This preserves only the geometric layouts while discarding all magnitude and sign information about $\{a_{I_i}\}$ (except for the fact that magnitudes were used to determine which terms to retain).

3. **Magnitude-flattened**:

$$m_K = \sum_{i=1}^{K} \frac{a_{I_i}}{|a_{I_i}|} \chi_{I_i} = \sum_{i=1}^{K} s_{I_i} \chi_{I_i}, \tag{19}$$

where we preserve the signs $s_{I_i} = \pm 1$ of the original coefficients but flatten their magnitudes. This allows us to investigate whether the "secondary" sign structure in Fourier space affects learnability.

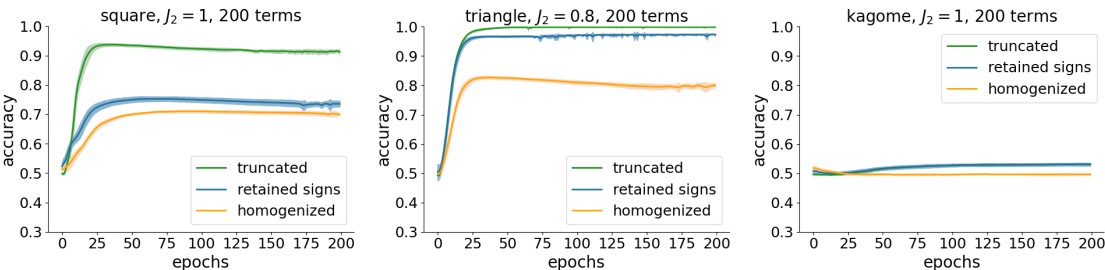

Figure 6: Accuracy of learning of sign structures obtained from ground states of the three considered models as a function of the number of epochs. Curves corresponding to simply truncated, homogenized, and sign-retained (flattened, see Sec. 5.3) sign structures are shown.

We train dense feedforward networks (one hidden layer, 512 neurons) on all three families using training sets of $\epsilon_{\text{train}} \cdot C_{N/2}^N$ vectors from the sector of zero magnetization, using exactly the same protocol as described above, though now with varying number of epochs.

The training results are shown in Fig. 6: green curves for simply truncated, orange for homogenized, and blue for flattened structures. The results reveal not only the importance of geometric structure but also the surprising role of coefficient signs in the Fourier domain, as we discuss below.

Focusing first on the simply truncated and homogenized structures (green and orange curves in Fig. 6), we observe clear differences in learnability. For the kagome lattice, both sign structures prove unlearnable by simple neural networks, as expected. For the less frustrated triangular and square lattices, the exact (simply truncated) sign structures are easier to learn than the homogenized ones. This is predictable since homogenization discards the benefit of decreasing coefficients $a_{I_i}$, making all parity functions equally important.

The next step is to study how learning complexity depends on the number of terms $K$ kept in the truncated series. In Fig. 7, we show neural network training results after a fixed number of epochs as a function of $K$. We chose 50 epochs as the point where learning of complete sign structures (with $K = C_{N/2}^N$) approaches a plateau without entering the overfitting regime. For each lattice, sign structure family, and value of $K$, we randomly initialized and trained the same neural network 8 times, then averaged the results.

The main counter-intuitive feature observed across all three quantum models is the highly non-monotonic dependence of accuracy on the number of parity functions (Fig. 7). For example:

- For the $J_2 = J_1 = 1$ square lattice, learning the homogenized sign structure with $K = 400$ parity functions is much easier than learning its substructure with only $K = 100$ parity functions.

- Learning a sign structure from the first $K \approx 40$ parity functions in the kagome case is hopeless, while learning from the first $K = 500$ parity functions of the triangular lattice Fourier expansion is straightforward.

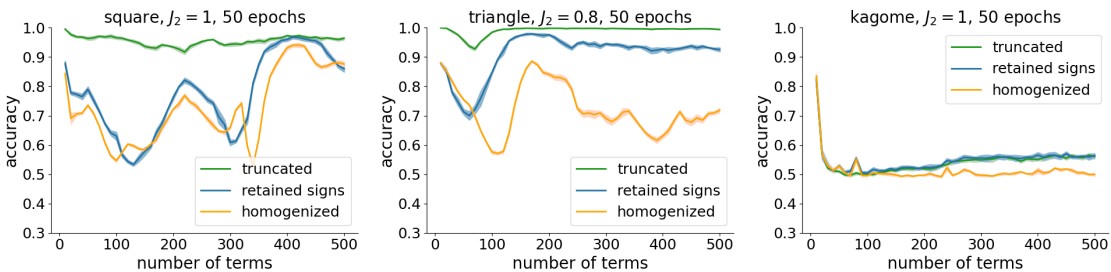

Figure 7: Accuracy of learning of sign structures obtained from ground states of the three considered models as a function of the number of kept expansion terms. For each sign structure, the learning has been performed 8 times, and the resulting curves represent averaging of the runs. Shades show (rather low) standard deviation over different runs.

- Even for the kagome lattice, there is a statistically significant peak around $K \approx 80$ parity functions, where accuracy is higher than for shorter expansions with $K \sim 20 - -50$.

This non-monotonic behavior suggests that geometric arrangements matter crucially. Depending on their spatial layout, two parity functions $\chi_{I_a}$ and $\chi_{I_b}$ may compete to different degrees. If they encode similar sign patterns, learning $\chi_{I_a} + \chi_{I_b}$ should be straightforward. However, if they give contradictory signals, learning their combination becomes more challenging.

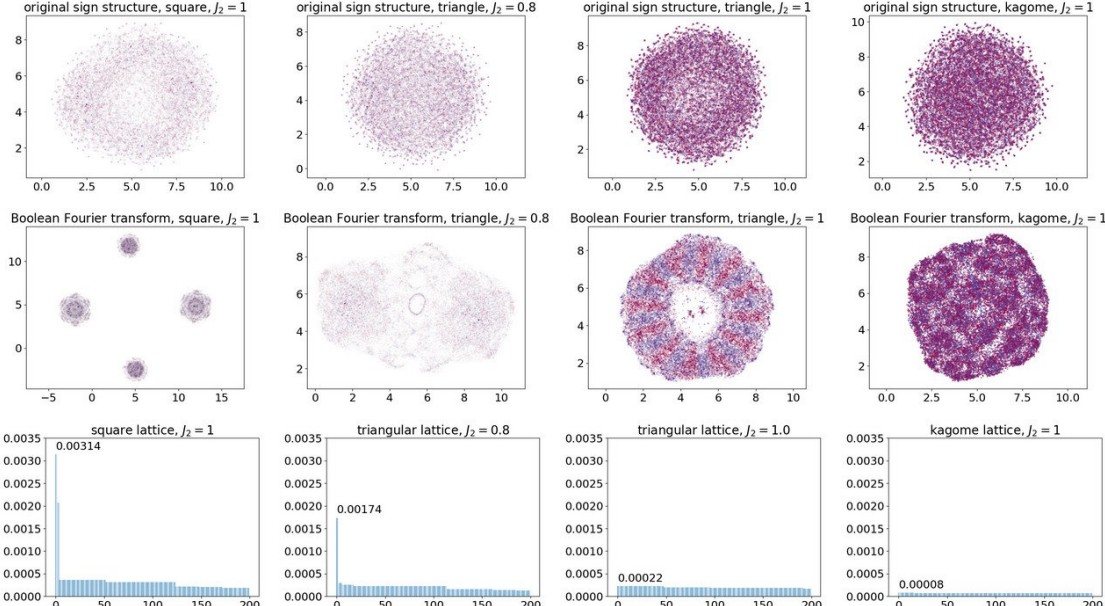

Figure 8: UMAP dimensional reduction of first 30000 expansion coefficients of the triangular lattice Heisenberg AFM sign structure in the original (left) and the Fourier (right) basis.

Having examined the effects of geometric structure through homogenization, we now investigate whether the signs of the Fourier coefficients themselves play a role in learnability. The magnitude-flattened (also referred to as 'flattened') structures $m_K$ (third type

introduced above) retain the signs $s_{I_i} = \pm 1$ of the original coefficients while setting all magnitudes equal, Eq. (19). Naively, there is no reason to expect that the signs $s_{I_i}$ should affect learning complexity compared to homogenized structures $h_K$, since the absolute strength of each term is identical in both cases. However, the blue curves in Figs. 6 and 7 reveal a systematic difference: learning flattened sign structures is consistently easier than learning homogenized ones. This surprising result indicates that the ground state sign structure possesses its own "secondary" sign structure in Fourier space, which positively affects learnability. The effect is most pronounced for the triangular lattice (intermediate complexity) and more marginal for the square and kagome lattices. In other words, not only do the geometric arrangements of parity functions matter, but the pattern of positive and negative coefficients in the Fourier expansion also encodes learnable structure.

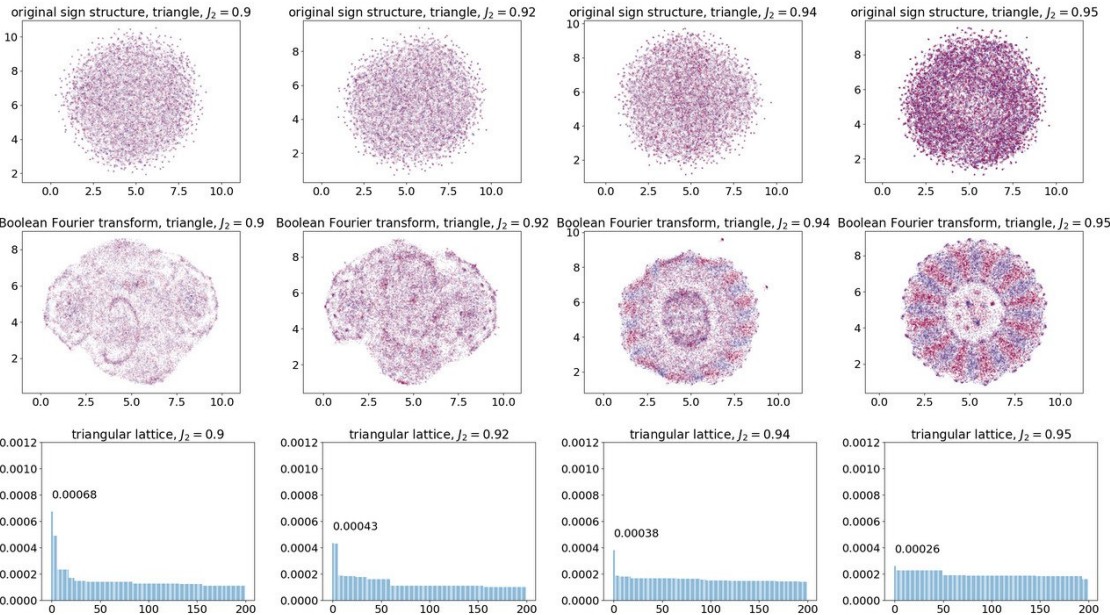

Figure 9: UMAP dimensional reduction of first 30000 expansion coefficients of the triangular lattice Heisenberg AFM sign structure in the original (left) and the Fourier (right) basis.

To better understand this secondary sign structure, we visualize both the original sign structure and its Fourier counterpart using Uniform Manifold Approximation and Projection (UMAP) [31]. This procedure projects each basis vector (represented as a binary sequence of length NN N) onto a two-dimensional plane while approximately preserving Hamming distances between them. While such projections are never exact, they can reveal hidden structures in the data.

In Fig. 8, we show UMAP projections for both the original computational basis and the Fourier basis across several cases: square lattice ($J_2/J_1 = 1$), triangular lattice ($J_2/J_1 = 0.8$ and $J_2/J_1 = 1$), and kagome lattice ($J_2/J_1 = 1$). Each point represents a basis vector, with color (red or blue) indicating the corresponding sign and transparency representing the coefficient amplitude, with more transparent points corresponding to smaller amplitudes.

The results are striking. In the original computational basis, the distribution of positive and negative signs appears too complex for UMAP to identify any underlying order, even for the square lattice where only four amplitudes have significant values. In contrast, the Fourier basis reveals much clearer structure: for square and triangular lattices, UMAP

either successfully separates positive and negative clusters (as in the $J_2/J_1 = 1$ triangular case) or at least shows clear attempts to organize the data. This visualization confirms that the Fourier transformation indeed creates a more structured representation of the sign patterns.

This geometric insight motivates the learning algorithms we develop in the following section, which exploit the structural properties of sign distributions in Fourier space to learn original sign structures without neural networks.

On a side note, we want to highlight a peculiar observation: the dramatic simplification of secondary sign structure separability in the triangular lattice case. At $J_2/J_1 = 0.95$, UMAP cleanly separates positive and negative vectors (Fig. 9), and this occurs precisely at the same $J_2$ value where both expansion-length-based complexity measures (Fig. 2) and learning complexity (Fig. 2 in [20]) are maximal. Remarkably, this transition happens before the system enters the spin-liquid state (around $J_2/J_1 = 1.25$) and appears to represent a hidden phase transition invisible to conventional observables.

## 3 New learning algorithms

So far, we used Boolean Fourier analysis to introduce complexity measures for sign structures and discovered key structural properties that determine learnability. Beyond defining measures based on series length, we found that geometric arrangements of parity functions matter. We also discovered that sign structures have "secondary" sign patterns in Fourier space, and this distribution of positive and negative coefficients in the Fourier expansion can be more organized than in the original computational basis (as mainly evident in the case of triangular lattice).

It can be shown that the Boolean Fourier transform has a natural quantum mechanical interpretation: transforming a signal to the Fourier basis means applying the Hadamard transform to it[4]. The fact that secondary sign patterns can become clearer in this basis suggests the Hadamard transform somehow "knows" about hidden patterns in the original sign structure. This leads to an obvious question: if the Hadamard transform reveals hidden structure, can we use this for learning? Though it is more intuition than theory, it suggests trying the Hadamard transform in supervised learning settings to see if it helps reconstruct sign structures from limited training data.

In this section, we show this intuition works. We develop two supervised learning algorithms that exploit the Boolean Fourier analysis to reconstruct complete sign structures from small training sets. Both drastically outperform neural networks in the frustrated regime and are based on simple explicit equations rather than opaque gradient-based optimization. These algorithms are tailored to supervised scenarios where we have access to exact signs on small subsets of the Hilbert space. While they cannot be used directly as variational ansätze due to computational constraints, their success suggests frustrated sign structures may not be as fundamentally complex as we thought.

Finally, on a practical side, we demonstrate that one can get best of both worlds and improve NQS variational ansätze using the ideas from Boolean Fourier analysis.

To evaluate these algorithms, we use an experimental setting similar to [20].

Our main testbed is 24-spin Kagome lattice, known to be the most challenging for neural networks in the frustrated regime ($J_2/J_1 > 0.51$). We compare the performance of our algorithms with the performance of a simple NQS. Exactly the same network architecture and learning protocol as outlined in Section 2.2 is used. The training goes for

---

[4]Note, that, in the context of ground states, we act with Hadamard operators on the sign structure, and not on the complete wave function. It will be explained in more detail soon.

1000 epochs (Adam optimizer, learning rate $10^{-3}$). To assess generalization, we evaluate predicted sign structures on a test set of 50.000 configurations sampled uniformly from the rest of the zero magnetization sector.

## 3.1 Hadamard learning

The key insight is that spin configurations and parity functions have a natural one-to-one correspondence: each spin configuration is characterized by which sites have spin-up, and each parity function $\chi_I$ is similarly characterized by a subset of sites $I$. This correspondence allows us to treat the Boolean Fourier transform as a linear operator on the spin system Hilbert space.

We can view any real-valued wavefunction $\psi$ as a signal on the Boolean hypercube. The Boolean Fourier transform (6) gives us coefficients $a_I$ for all spin subsets $I$. We can then define a new wavefunction $\tilde{\psi}$ by assigning these coefficients to spin configurations:

$$\langle \tilde{\psi} | s_\uparrow(I) \rangle = a_I, \tag{20}$$

where $s_\uparrow(I)$ is a spin configuration with spins up on sites in $I$. The transformation

$$T : T |\psi\rangle = |\tilde{\psi}\rangle \tag{21}$$

is the Hadamard transform [32], which applies a Hadamard gate to each spin. Worth noting that the Heisenberg Hamiltonian commutes with this operation, so any non-degenerate eigenstate remains invariant under the transform.

The learning algorithm based on the Hadamard transform comprises three steps:

1. Given a training set $\{\sigma_1, \ldots, \sigma_K\}$ with known signs, construct an auxiliary wavefunction:

$$|\psi_{\text{train}}\rangle = \sum_{i=1}^{K} |\sigma_i\rangle \, \text{sign}\langle \psi_{\text{GS}} | \sigma_i \rangle. \tag{22}$$

   This wavefunction has unit amplitudes at training configurations (with correct signs) and zero amplitudes elsewhere (at this point, we do not care about normalization).

2. Apply the Hadamard transform to spread the information localized on the training subset across the entire basis and obtain a *predictor* signal:

$$|\psi_{\text{predict}}\rangle = T|\psi_{\text{train}}\rangle \tag{23}$$

   Note that although the Hadamard transform formally changes the basis, here we regard the resulting wavefunction $|\psi_{\text{predict}}\rangle$ as written in the original basis.

3. Make predictions outside of the training subset by assigning signs of configurations $\sigma$ to be equal to $\text{sign}(\langle \sigma | \psi_{\text{predict}}\rangle)$. In doing that, we essentially conjecture that sign structures of $|\psi_{\text{GS}}\rangle$ and $|\psi_{\text{predict}}\rangle$ correlate.

In Fig. 10, we present results for the 24-spin Kagome lattice. One can see that Hadamard learning dramatically outperforms neural networks in the frustrated regime, as measured by sign overlap [9]:

$$\mathcal{O}_S = \frac{\sum\limits_{i \in \text{test set}} |\psi_{\text{GS}}^i|^2 \, \text{sign}(\langle \sigma_i | \psi_{\text{GS}}\rangle) \cdot \text{sign}(\langle \sigma_i | \psi_{\text{predict}}\rangle)}{\sum\limits_{i \in \text{test set}} |\psi_{\text{GS}}^i|^2} \leq 1 \tag{24}$$

Remarkably, Hadamard learning shows meaningful generalization even with $\varepsilon_{train} = 10^{-4}$, learning the sign structure from just 270 training examples[5]! However, in the unfrustrated regime, neural networks perform better if the training set is large enough ($\varepsilon_{\text{train}} \gtrsim 0.01$ in our case).

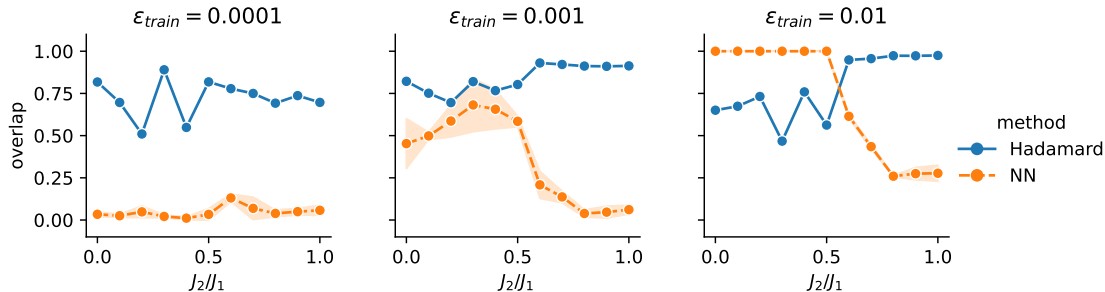

Figure 10: Comparison of the generalization abilities of the Hadamard learning and neural networks. Shaded regions are 95% percentile band.

## 3.2 Fourier learning

While Hadamard learning applies a single global transform to training data, we can also approach sign learning through Monte Carlo estimation of Fourier coefficients – a strategy conceptually closer to conventional neural quantum state training. Let us first outline the core idea of the algorithm, which we call Fourier learning, and then discuss the related nuances.

The first step is to estimate the Boolean Fourier coefficients $a_I$ from the training data and then reconstruct the complete sign structure. To do that, one can note that each coefficient in the Fourier expansion of signal $f$ can be written as:

$$a_I = \langle \chi_I | f \rangle = \langle \chi_I(\sigma) f(\sigma) \rangle_{\sigma \sim \text{Uniform}(B_n)} \equiv \sum_{j=1}^{2^N} \frac{1}{2^N} \chi_I(\sigma) f(\sigma), \tag{25}$$

i.e. as an expectation value of $\chi_I f$ function computed with respect to uniform distribution on the Boolean hypercube, $\text{Uniform}(B_n)$.

This means that each $a_I$ can be estimated through i.i.d. Monte Carlo sampling of spin configurations from $\text{Uniform}(B_n)$:

$$a_I \simeq \hat{a}_I \equiv \frac{1}{K} \sum_{j=1}^{K} \chi_I(\sigma^j) f(\sigma^j), \quad K \ll 2^N. \tag{26}$$

Once the Fourier coefficients are estimated, we reconstruct the sign structure as

$$f_{\text{pred}}(\sigma) = \text{sign}\left( \sum_{I \in \mathcal{R}} \hat{a}_I \chi_I(\sigma) \right), \tag{27}$$

where $\mathcal{R}$ (for "relevant") is the set of Fourier terms we decided to keep in our reconstruction.

---

[5]The total dimension of the sector is $D = \binom{24}{12} \simeq 2.7 \cdot 10^6$

We select $\mathcal{R}$ to contain the $M$ terms with the largest estimated Fourier coefficients $|\hat{a}_I|$, setting all other coefficients to zero, and assume that the corresponding parity functions $\chi_I$ are known. The coefficients $\hat{a}_I$ are all estimated from the same training set $\{(\sigma^j, f(\sigma^j)): j = 1, \ldots, K\}$, making it conceptually similar to the conventional supervised scheme.

The Fourier polynomial $\hat{f}$ constructed from these estimated coefficients approximates the true signal:

$$f \simeq \hat{f} = \sum_{I \in \mathcal{R}} \hat{a}_I \chi_I. \tag{28}$$

Since our Monte Carlo estimates can produce non-integer values while the true sign structure $f$ takes only values $\{-1, 0, 1\}$, we apply the sign function in (27) to recover a proper sign structure.

In practice, we do not know which terms are most relevant beforehand. The Fourier learning algorithm estimates all coefficients $\hat{a}_I$ from the training data, then keeps only the $M$ terms with largest absolute values.

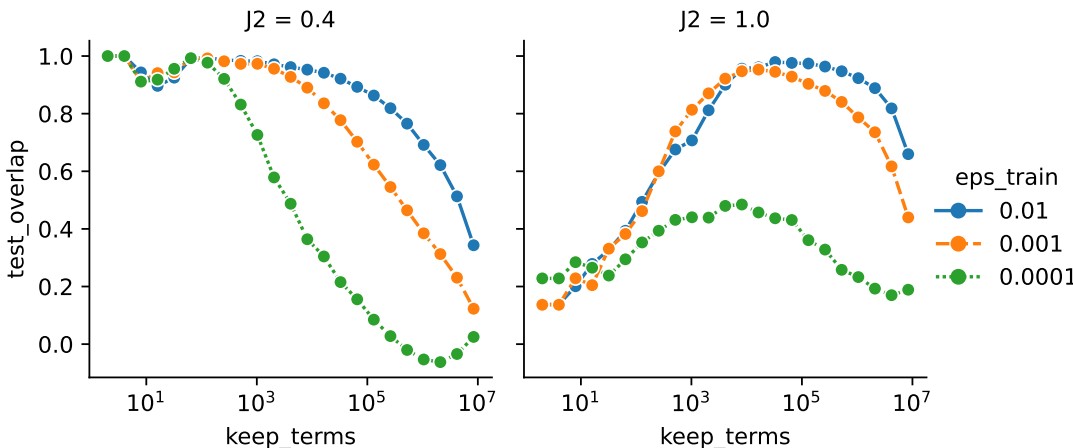

Figure 11: The dependence of the generalization ability of the truncation of the learned Fourier polynomial on the number of terms to keep ($M$). Left: unfrustrated system, right: frustrated system.

Two notes should be added. First, simply increasing $M$ does not always improve performance, see Fig. 11. Moreover, if we keep all $M = 2^N$ terms, we are effectively reconstructing a function that matches the training data exactly (up to normalization) but is zero everywhere else – perfect overfitting with no generalization:

$$\tilde{f}(\sigma^l) = \sum_I \hat{a}_I \chi_I(\sigma^l) = \sum_I \frac{1}{K} \sum_{j=1}^K f(\sigma^j) \chi_I(\sigma^j) \chi_I(\sigma^l) = \tag{29}$$

$$\frac{1}{K} \sum_{j=1}^K f(\sigma^j) \sum_I \chi_I(\sigma^j) \chi_I(\sigma^l) = \frac{2^N}{K} \sum_{j=1}^K f(\sigma^j) \delta_{jl} \sim f(\sigma^l) \text{ for } 1 \le l \le K,$$

if summation goes over all subsets $I$. Here we used the property of parity functions:

$$\sum_I \chi_I(\sigma^j) \chi_I(\sigma) = \begin{cases} 2^N \text{ if } \sigma = \sigma^j, \\ 0 \text{ if } \sigma \neq \sigma^j. \end{cases} \tag{30}$$

The algorithm also naturally handles non-uniform sampling. If we sample configurations with probability $p(\sigma)$, we effectively estimate coefficients for the rescaled signal $f(\sigma)p(\sigma)$. Since applying sign preserves the sign structure regardless of the scaling, this gives us flexibility in choosing sampling strategies. The truncation parameter $M$ and sampling distribution $p(\sigma)$ are the only hyperparameters of this algorithm.

We now demonstrate Fourier learning's superior generalization compared to neural networks, using the same experimental protocol. The main subtlety is selecting the truncation parameter $M$ without knowing the ground truth. Simply choosing the best-performing $M$ for each case would be unfair, as this information is not available in practice.

Instead, we find the most stable prediction across different truncation levels. We test values $M = 2^m$ for $7 \leq m \leq N$, computing the sign overlap between consecutive predictions $f_{\text{pred}}^m$ and $f_{\text{pred}}^{m+1}$. We select the $M$ that maximizes this overlap, corresponding to regions where the prediction stabilizes as we vary the number of kept terms. We exclude very small values $M < 100$ to avoid premature stabilization.

In Fig. 12, we show results for the 24-spin Kagome Heisenberg model. One can see that Fourier learning dramatically outperforms the neural network in both frustrated and unfrustrated regimes.

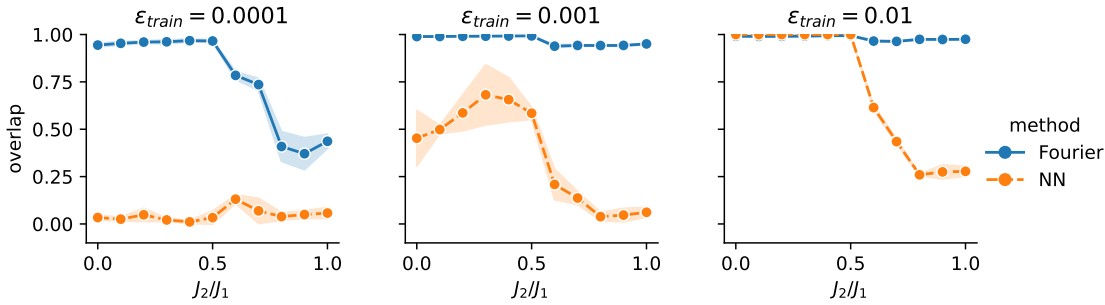

Figure 12: Comparison of the generalization abilities between Fourier learning and neural networks. Shaded regions are 95% percentile band.

Finally, we examine sensitivity to the sampling power $\alpha$ in $p(\sigma) = |\psi_{\text{GS}}(\sigma)|^\alpha$. Figure 13 reveals several findings. Uniform sampling ($\alpha = 0$) produces nearly useless predictions for both Fourier learning and neural networks, confirming that amplitude-weighted sampling is crucial. Interestingly, for very small training sets ($\varepsilon_{\text{train}} = 10^{-4}$), the optimal $\alpha$ exceeds 2, suggesting we should oversample high-amplitude regions when data is scarce.

Most remarkably, Fourier learning achieves overlap $> 0.95$ with just $\varepsilon_{\text{train}} = 10^{-4}$ - learning from only 270 configurations out of 2.7 million. Neural networks struggle to learn anything meaningful even with $\varepsilon_{\text{train}} = 0.01$.

The consistently low prediction accuracy (fraction of correctly guessed signs) compared to sign overlap reflects our sampling strategy: by focusing on high-amplitude regions with $\alpha > 0$, we extract more information where it matters most for the overlap metric. The poor performance of all methods under uniform sampling ($\alpha = 0$) suggests that learnable sign structure primarily exists in high-amplitude regions of the Hilbert space.

We have introduced Hadamard and Fourier learning as explicit algorithms for supervised sign structure reconstruction. Both dramatically outperform neural networks in this task, but their practical integration with variational methods requires consideration.

The critical issue is that conventional variational optimization is unsupervised: we optimize energies without knowing signs or amplitudes on any subset of configurations.

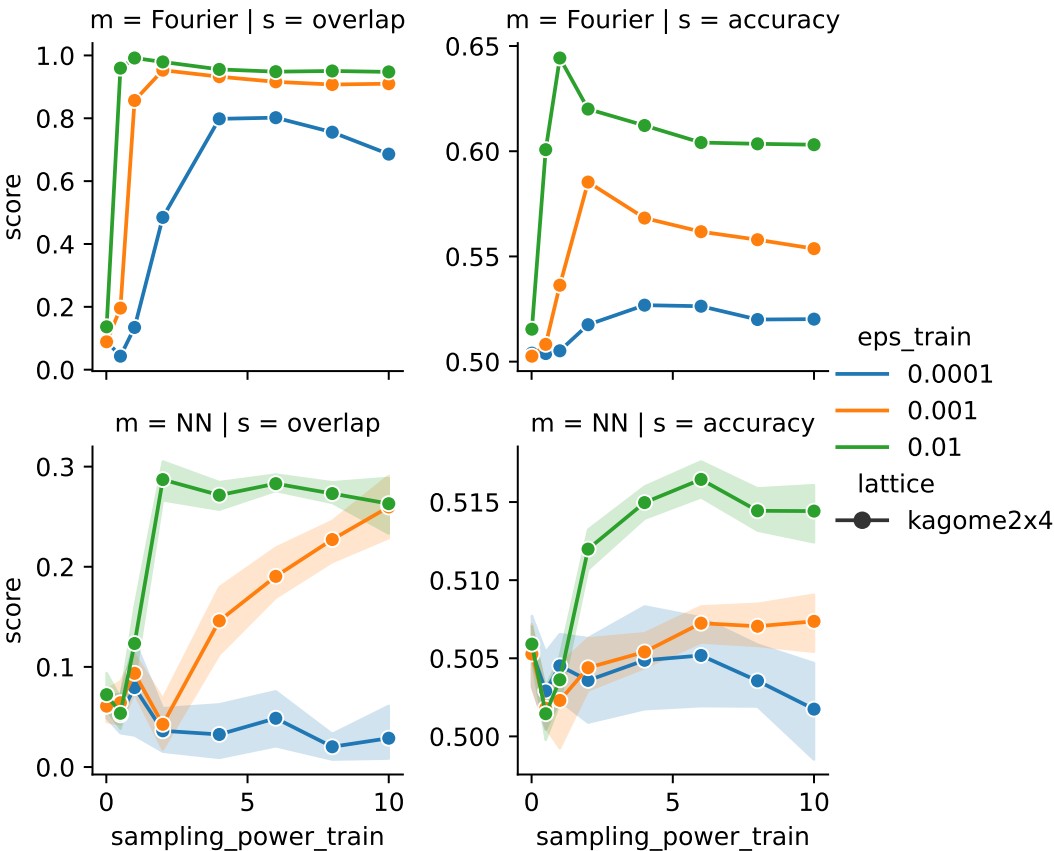

Figure 13: The best score value for various sampling powers. Lattice is 24-spin Kagome and $J_2/J_1 = 1$. Horizontal axis is sampling power $\alpha$. Top row: Fourier learning, bottom row: Neural Network. Left column: score is sign overlap. Right column: score is accuracy. For accuracy, level 0.5 corresponds to absence of any correlation between the prediction and the ground truth.

However, the Supervised Wavefunction Optimization (SWO) framework [33] provides a potential bridge between our methods and variational approaches. While we refer readers to the original paper for full details, SWO essentially performs iterative supervised updates of the neural network, with each step maximizing the overlap between the NQS and an explicit intermediate target that can be represented directly. Hadamard learning could naturally integrate into this framework, offering a principled way to incorporate Fourier insights into variational optimization.

Fourier learning faces additional computational constraints since determining the optimal truncation $M$ requires estimating all $2^N$ coefficients. However, both approaches demonstrate that frustrated sign structures contain more learnable structure than previously thought. This suggests new directions for improving neural quantum states by incorporating Boolean Fourier analysis insights. We explore this possibility in the next section.

### 3.3   Improving neural networks by parity augmentation

While Hadamard and Fourier learning cannot yet serve as drop-in replacements for variational methods, their success hints that Boolean Fourier analysis reveals something important about quantum sign structures. We now show how to incorporate these ideas into conventional neural quantum states through parity augmentation by adding selected parity functions as additional network inputs alongside spin configurations. This represents a form of feature engineering that works in any training context. Here we stick to the supervised learning scheme, but the same augmentation can be incorporated in variational energy minimization schemes.

We tested three strategies for selecting which parity functions to include as additional features:

1. Uniform sampling from the complete set of parity functions

2. Uniform sampling of parity functions of a specific degree (number of spins involved)

3. Weighted sampling with probability proportional to some power of the parity function Fourier coefficient in the true sign structure Fourier series.

In each case, we sample up to 8 parity functions to augment the neural network input.

Table 1 shows results for the 24-spin Kagome lattice at $J_2/J_1 = 1$. The weighted sampling strategy (strategy 3) works best, significantly outperforming the baseline neural network with sign overlap $> 0.7$ compared to $\approx 0.26$. This confirms that including parity functions with large Fourier coefficients (i.e. those most correlated with the true sign structure) indeed helps the network learn. The degree-based results reveal that low-degree parities (degree 2) actually hurt performance compared to the baseline, while higher degrees perform better. Since dominant Fourier terms in frustrated Kagome systems typically have degree around 8, we tested random sampling at this degree. It outperforms degree 2 but falls short of the weighted strategy. Degree 12 works slightly better than degree 8. Interestingly, uniform unconstrained sampling performs similarly to degree 8 selection.

Generally, we see that by adding several parity functions of large degree one usually improves the performance of the model, but too many of them can lead to slight overfitting.

| sampling type | parity degree | sampling power | # parities | test overlap mean | sem |
|---|---|---|---|---|---|
| Baseline | | | | 0.2602 | 0.0170 |
| Unconstrained | | | 1 | 0.4451 | 0.0406 |
| | | | 2 | 0.4463 | 0.0292 |
| | | | 4 | 0.4751 | 0.0171 |
| | | | 8 | **0.4874** | 0.0194 |
| Degree-specific | 2 | | 1 | 0.2158 | 0.0108 |
| | 2 | | 2 | 0.2308 | 0.0071 |
| | 2 | | 4 | 0.1958 | 0.0066 |
| | 2 | | 8 | 0.1255 | 0.0108 |
| | 8 | | 1 | 0.3463 | 0.0173 |
| | 8 | | 2 | 0.4423 | 0.0166 |
| | 8 | | 4 | 0.4346 | 0.0168 |
| | 8 | | 8 | 0.3976 | 0.0174 |
| | 12 | | 1 | 0.4074 | 0.0303 |
| | 12 | | 2 | 0.5165 | 0.0301 |
| | 12 | | 4 | **0.5186** | 0.0160 |
| | 12 | | 8 | 0.4738 | 0.0193 |
| Weighted | | 2 | 1 | 0.4472 | 0.0441 |
| | | 2 | 2 | 0.5591 | 0.0301 |
| | | 2 | 4 | 0.6016 | 0.0078 |
| | | 2 | 8 | 0.5760 | 0.0188 |
| | | 4 | 4 | 0.6862 | 0.0058 |
| | | 4 | 8 | 0.6911 | 0.0064 |
| | | 8 | 4 | **0.7194** | 0.0064 |
| | | 8 | 8 | 0.7134 | 0.0052 |

Table 1: Comparison between different augmentation schemes.

# 4 Discussion

We have performed an extensive analysis of ground state sign structure complexity in several quantum many-body models using Boolean Fourier analysis of binary signals on hypercubes. This approach was motivated by two key observations: parity functions (equivalently, XOR logical functions) represent the minimal primitive producing linearly non-separable data and can be regarded as fundamental units of complexity, while the Marshall sign rule of bipartite antiferromagnets can be represented with a single parity function, making it the simplest nontrivial quantum sign structure. Our main goal was to introduce a universal formalism for assessing sign structure complexity that is independent of specific variational ansätze or optimization algorithms.

Given that sign structure complexity is already significant for small systems, we studied ground states of 24-spin Heisenberg models on square, triangular, and Kagome lattices with varying frustration levels controlled by $J_2/J_1$. We investigated how the learning complexity of these sign structures, which is defined as the ability of neural networks to predict signs across the Hilbert space from limited training data, relates to the properties of their Boolean Fourier expansions.

We found that the number of relevant Fourier terms generally correlates with learning complexity, with highly frustrated quantum states exhibiting longer heavy-tailed expansions. However, this relationship is not the complete story. Through systematic studies of truncated ground state sign structures and artificially constructed signals, we identified non-monotonic regimes where neural network learning ability depends unexpectedly on descriptive complexity: shorter binary signals can sometimes be more difficult to learn than longer ones, even when the longer signal contains the shorter one as a subset.

Since Fourier coefficients have their own signs, we can define "secondary" sign structures in the parity function basis, which have rather distinct properties. Experiments with artificially constructed signals showed that signals incorporating information about Fourier coefficient signs are systematically easier to learn than those excluding this information, even though both should naively have the same degree of linear non-separability. Furthermore, UMAP dimensional reduction showed that secondary (Fourier) sign structures are typically more organized than original ones, and the algorithm successfully separates or attempts to separate clusters of positive and negative signs, while original sign structures often appear to lack intrinsic order.

Finally, motivated by these insights, we developed two algorithms – Hadamard and Fourier learning – that drastically outperform neural networks in supervised sign learning. Both procedures exploit the fact that the Boolean Fourier transform corresponds to applying Hadamard gates simultaneously to all spins, effectively spreading information from a small set of training configurations across the entire Hilbert space. We attribute their remarkable accuracy to the invariance of the studied quantum systems under the Hadamard transform.

The Hadamard learning algorithm has practical potential. While Fourier learning faces scalability issues due to exponential coefficient estimation requirements, Hadamard learning could be integrated into the Supervised Wavefunction Optimization (SWO) framework for realistic variational ground state calculations. The practical issue here is obtaining the initial "seed" signs on a subset of configurations, as required by Hadamard learning. Here, a connection can be made with the approach that maps quantum sign structures onto classical Ising models [9]. Then, the required subset of signs can be bootstrapped from wavefunction amplitudes through combinatorial optimization, which naturally fits into the SWO iterative scheme. Given that Hadamard learning generalizes much better than neural networks and requires no iterative weight updates, this combined approach

could make SWO both more accurate and faster.

The parity function feature engineering we tested here can be readily transferred to conventional neural quantum state training based on energy optimization.

Finally, our findings point to interesting connections with quantum computing. The fact that Boolean Fourier transforms correspond to single layers of Hadamard gates applied to auxiliary wavefunctions calls for exploring hybrid quantum-classical approaches. While previous work showed that some quantum states require deep circuits for sign positivization [34], what we show here is that by representing the sign structure as an auxiliary wavefunction (which is not invariant under the Hadamard transform), a single layer of Hadamard gates can make signs more structured and learnable, as we demonstrated for the triangular lattice Heisenberg model. This suggests that combining simple quantum circuits with neural networks could be a useful approach for studying quantum systems with complex sign structures.

**Acknowledgments** I.S. is grateful to Vladimir Podolskii for an inspiring talk on application of Boolean Fourier analysis to machine learning problems and further valuable discussions.

**Funding information** I.S., M.I.K., and T.W. acknowledge the support received from NWO via Spinoza prize of M.I.K. The work of A.A.B. was supported by NWO grant OCENW.M.23.044. The work of A.K. was supported by the Interdisciplinary Research Platform grant from Radboud University.

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

## A  Visualization of parity functions

Here, we show the first few parity functions that have largest coefficients $a_I$ in the Fourier expansions of the corresponding sign structures.

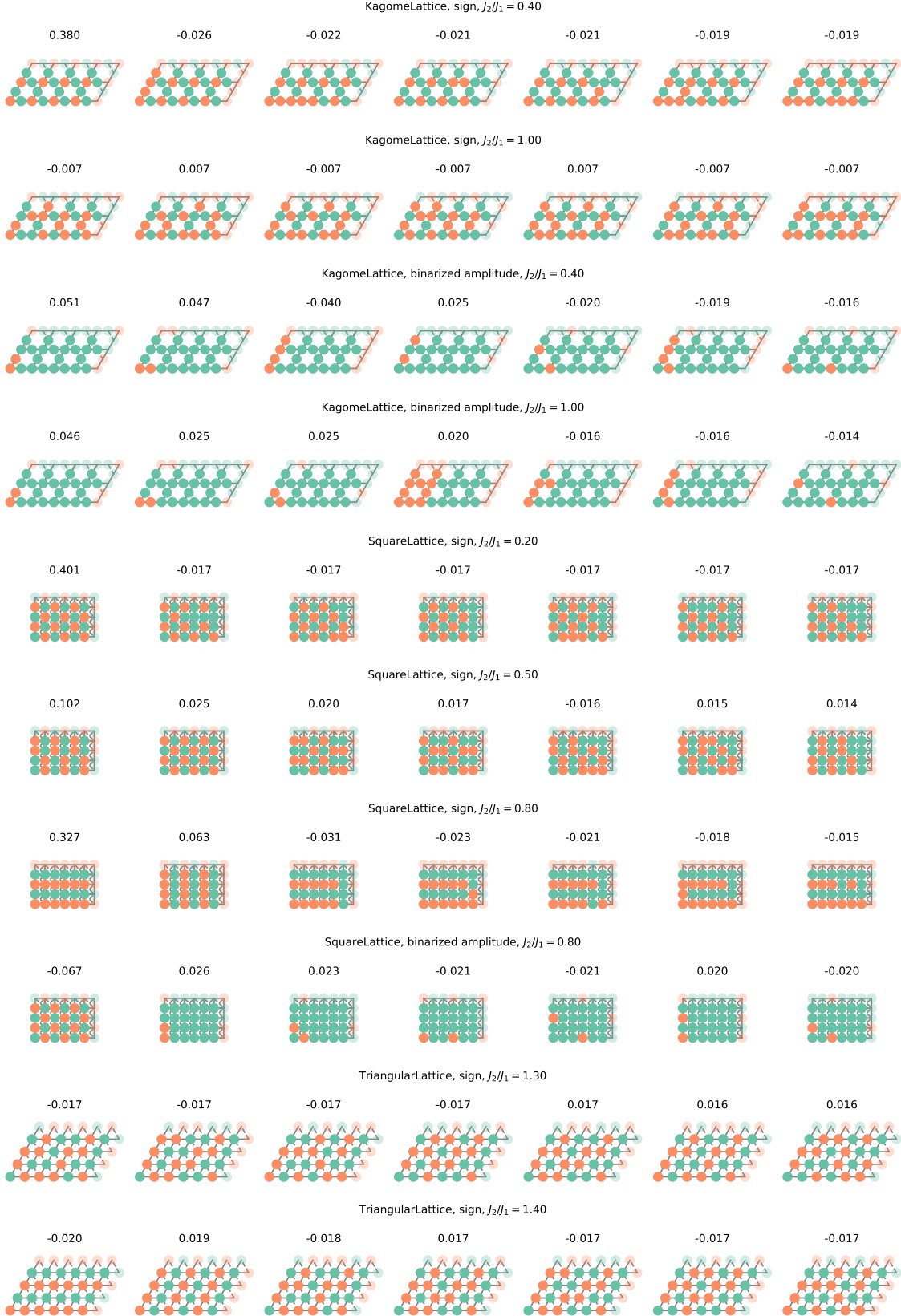

Figure 14: Visualization of Fourier expansions. Each panel represents one parity function, up to symmetries of the lattice and sign flips. The label is the corresponding coefficient. Highlighted sites are those included in the parity function.