# Peer review of "Learning complexity of many-body quantum sign structures through the lens of Boolean Fourier analysis"

_SciPost Physics_

## Round 1 · Referee Report · Anonymous (Referee 1) · 2025-11-10

Disclosure of Generative AI use

The referee discloses that the following generative AI tools have been used in the preparation of this report:

For the improvement of the text presentation.

Strengths

Important, relevant, and timely topic. Presentation very good.

Weaknesses

None.

Report

In this work the authors introduce Boolean Fourier analysis as a novel framework for analyzing the complexity of sign structures in quantum many-body systems, particularly frustrated antiferromagnetic Heisenberg models. Representing sign structures as Boolean functions on N-dimensional hypercubes, the authors examine both description and learning complexities. They develop different complexity measures and establish correlations between reconstruction and learning complexities. This work reveals that secondary sign structures in Fourier space exhibit greater organization than in the computational basis. Building on this insight, the authors develop supervised learning algorithms (Hadamard and Fourier learning) that significantly outperform conventional strategies in neural quantum states, particularly for frustrated systems with limited training data. The work also demonstrates the potential of feature engineering through parity augmentation to enhance neural quantum state approximations. Additionally, connections to quantum computing are established, where Boolean Fourier transforms correspond to Hadamard gate operations. This work provides insights into the relationship between mathematical representation and computational learnability of quantum states, with implications for variational quantum state methods in strongly correlated systems.

The paper is well written and straightforward to follow. It covers an important, relevant, and timely topic related to the emergent numerical method of neural quantum states. The results will be very useful to the community of neural quantum states, and potentially beyond to understand the sign structures of quantum wave functions.

This paper is publishable in SciPost in its current form.

Requested changes

None.

Recommendation

Publish (surpasses expectations and criteria for this Journal; among top 10%)

---

## Round 1 · Referee Report · Anonymous (Referee 2) · 2025-11-14

Strengths

original, insightful

Weaknesses

1-Analysis limited to small systems
2-No clear scaling study, neither in terms of system size nor in terms of number of parameters to achieve a given accuracy
4-Missing methodological details

Report

The authors address the complexity of describing and learning the sign structure of the ground state of frustrated quantum spin systems. Upon introducing concepts from Boolean Fourier analysis, they construct a set of complexity metrics relying on the power spectrum of the ground state expressed in a computational basis.

Using these metrics, they investigate the sign-structure complexity of the ground-state wave function of several frustrated lattices, showing a remarkable overlap between high values of these metrics and the frustrated regimes, where usual Marshall-sign tricks fail. Interestingly, these Marshall signs appear to correspond to the leading Fourier amplitudes. Their calculations hint at an exponential growth of the number of Fourier components required to faithfully reconstruct the sign structure, although over the limited lattice-size range available within exact diagonalisation. Besides the size of the support of the power spectrum, the authors observe that the learnability is impacted by the actual relative Fourier amplitudes. Finally, the authors observe that the Boolean Fourier transform eases the clustering of signs.

The authors are interested in a rather peculiar supervised-learning protocol in which the sign of the ground-state wave function is known for a set of spin configurations. In this setting, they explore how a trained two-layer perceptron generalises to the signs of other amplitudes of the same many-body wave function. The authors propose two different heuristics to guess the sign structure from a set of examples and numerically show that they outperform the previous supervised-learning approach. However, in principle, these entail a complexity scaling exponentially. Finally, they test augmenting the input of the two-layer feedforward network with parity functions.

Overall, I believe the topic is timely and that a revised manuscript may make a valuable contribution to the field. However, I would like the authors to address the points given hereafter before I can issue any definite recommendation.

Requested changes

1-The supervised-learning setting is highly not standard in the field of neural-network approaches to ground-state quantum physics and should be made more explicit since the first numerical examples. I found it quite hard to understand at the beginning and it is not always obvious that the same protocol is reused in the following sections. I think it should be formally introduced in some methodological section/subsection that the authors can refer throughout the manuscript. Unless I am mistaken, $\varepsilon$ is not really explicitly defined.
2-I believe that at some points of the manuscript, readers might confuse your supervised training and the standard variational Monte Carlo. For instance, from the abstract: “(…) demonstrate that such polynomials can potentially serve as *variational* ansätze for the complex sign structures that dramatically outperform *neural networks* in terms of generalization ability.” It could seem like the authors do variational simulation, but they actually never do (?)
3-Minor but in 2.2, maybe bias could be called inductive bias, to distinguish it from the Monte Carlo bias, more commonly referred to as bias by researchers in neural quantum states, if this is a community the authors want to address.
4-Staying on the bias aspect, I understand the authors want to capture a general description complexity measure and thus choose a network with no particular structure, but as they clearly outline in the introduction such a general measure is ill-defined. A different architecture, though generic, might yield a completely different result; and it is indeed the case: transformers seem to learn the sign structure of frustrated spin systems with minimal physical input (see e.g. 10.1103/PhysRevResearch.6.043280 or 10.1038/s42005-024-01732-4). Therefore, maybe one should not motivate it from an inductive bias standpoint, which is fine: feedforward networks are still the most obvious test for feature extraction. Also it is clear how they infer the sign by learning a boundary in their feature space, so the amount of hidden nodes relates to the complexity of that boundary and hence is an interesting form of descriptive complexity.
5-Unless benchmarks are performed against the state-of-the-art architecture for a given system, the authors should avoid sentences like “we compare against neural networks” and instead specify “against feedforward neural networks”.
6-I do not understand the choice of test set. The reason one chooses a small test set in supervised learning is that one has limited data. Therefore, most of the available data is used for training, reserving a few for the testing. But here one has access to all the data and the authors work at fixed training-set size. Therefore, why not test on all configurations that were not part of the training set?
7-In the bottom panels of Fig. 9, what are the axes’ labels?
8-In Section 2, the authors write: “Remarkably, this transition happens before the system enters the spin-liquid state (around J2/J1 = 1.25) and appears to represent a hidden phase transition invisible to conventional observables.” For these very small lattice sizes, there could be significant finite-size effects that shift and enlarge the critical-point region. In the absence of any finite-size scaling study, I do not think their assertion is justified.
9-In Eq. (25), missing j in the summand (?)
10-I do not think it is reasonable to suggest Monte Carlo sampling from the uniform distribution. It is not only that the number of samples will scale exponentially, there is no way to likely get a better result than the exact sum of Eq. (25) at a lower computational cost, except from being accidentally lucky. I really do not think such a proposal can be written in an accepted manuscript.
11-Can’t one simply sample $\sigma$ from $f^2$ and use $\mathbb{E}[\xi_\mathrm{loc}(\sigma)]$, with a local estimator $\xi_\mathrm{loc}(\sigma) := \xi_I(\sigma)/f(\sigma)$?
12-In Fourier learning, you still need, in principle, to obtain exponentially many terms of the power spectrum. Isn’t that the complexity of getting the exact ground state?
13-I had a hard time understanding what f corresponds to, e.g. in Eq. (29). Does it relate to $\psi_\mathrm{train}$?
14-In Fig. 11, how is the trend observed compatible with Eq. (28)?
15-In 3.3, honestly any uniform sampling should be removed.
16-In 3.3, please show explicitly how the parity functions are inputted in the network. It is not so clear without an equation or going to check the code.
17-Open question: is there ultimately any algorithm in Section 3 of subexponential complexity that is guaranteed to improve the learning of the sign-structure in a scalable way?
18-Open question: Suppose one can target a few leading terms of the Fourier expansion. Could one use them within a VMC pipeline to improve the variational energy, much like it is done with the usual Marshall sign? Have the authors tried this?
19-Open question: The usual quantum systems with complicated sign structures are frustrated magnetic systems and fermions. Have the authors ever attempted a similar study as in Fig. 4 for fermions in second quantisation? For instance, free fermions on a lattice of dimension d>1 can be captured exactly by a single Slater determinant (polynomially many parameters) whereas a feedforward network requires exponentially many parameters to learn the sign structure stemming from fermionic statistics, even though the state is “trivial” (see for instance 10.1103/PhysRevLett.134.079701). For instance, it would be interesting to understand how the complexity of the sign structure is decreased upon using as a prior the signs of the determinant (known a priori, essentially the parity of the permutation of the configuration to bring it into normal ordering); like what is done in 10.1103/PhysRevResearch.3.043126.

Recommendation

Ask for major revision

---

## Round 1 · Referee Report · Anonymous (Referee 3) · 2025-11-17

Strengths

1 - Original approach to investigate the sign structure of a quantum wave function
2 - Potential applications to improve neural quantum states (NQS) for frustrated magnets

Weaknesses

1 - This study is limited to small systems accessible by Exact Diagonalization (ED), namely N=24.
2 - Some learning schemes cannot be easily generalized to larger systems.

Report

The authors provide a novel indicator to measure the sign structure of the groundstate of a frustrated spin model, based on Boolean Fourier analysis. They introduce several measures for this complexity, all behaving similarly. For instance, having few vs many coefficients in the Fourier decomposition. Indeed, in the simplest nonfrustrated case (on bipartite lattices), there is a single coefficient, which can be deduced from the Marshall-Peierls sign rule.

Intuitively, the complexity should be related to the learning ability. However, they also provide some detailed analysis with counterintuitive results: on the triangular lattice, the accuracy is non-monotonic in so-called learnability. This is argued to be due to some specific structure of the parity functions. This result itself deserves publication in my opinion since it provides a deeper understanding of the sign structure.

The last part provides two algorithms proposed to be superior to traditional neural networks (NN). This section is less concinving since (i) Fourier learning requires to compute an exponentially large number of coefficients; (ii) comparison is made to single-layer feedforward NN, which is not the state-of-the-art achitecture for frustrated magnets.

In conclusion, I recommend the paper for publication as it provides some novel insight into the sign structure of quantum wavefunction, but I ask the authors to answer some comments first.

Requested changes

1 - The notations are a bit misleading since J1-J2 models have been studied extensively in quantum magnetism, J1 being the nearest (and J2 the next-nearest) neighbor bonds. Here the authors use J1 and J2 in a different sense, which could cause some confusion. I would recommend to add a warning about this notation.
2 - Regarding the J1-J2 square lattice, I would recommend to add more recent references (e.g. PhysRevLett.121.107202; Science Bulletin 67, 1034 etc.) discussing its phase diagram.
3 - Since the Fourier learning algorithm cannot be used for larger sizes N, this could be removed from the main text and put in Appendix. Regarding Hadamard learning, if this can be used for larger N, it would be quite useful to provide such results.
4 - Using lattice symmetries, it is possible to study quite larger systems with ED (N=50 has been done using massively parallel ED code but N=36 could be done straightforwardly). Why did you limit your analysis to N=24 ?
5 - Would there be some possibility to generalize this approach to intrisically complex wavefunctions (e.g. when the Hamiltonian has complex terms) ?

Recommendation

Ask for minor revision

---

## Editorial Decision

awaiting_resubmission